# Enhancing Trauma Care: Machine Learning-Based Photoplethysmography Analysis for Estimating Blood Volume During Hemorrhage and Resuscitation

**DOI:** 10.3390/bioengineering12080833

**Published:** 2025-07-31

**Authors:** Jose M. Gonzalez, Lawrence Holland, Sofia I. Hernandez Torres, John G. Arrington, Tina M. Rodgers, Eric J. Snider

**Affiliations:** U.S. Army Institute of Surgical Research, JBSA Fort Sam Houston, San Antonio, TX 78234, USA

**Keywords:** machine learning, feature extraction, predictive modeling, hemorrhage, resuscitation, shock, trauma

## Abstract

Hemorrhage is the leading cause of preventable death in trauma care, requiring rapid and accurate detection to guide effective interventions. Hemorrhagic shock can be masked by underlying compensatory mechanisms, which may lead to delayed decision-making that can compromise casualty care. In this proof-of-concept study, we aimed to develop and evaluate machine learning models to predict Percent Estimated Blood Loss from a photoplethysmography waveform, offering non-invasive, field deployable solutions. Different model types were tuned and optimized using data captured during a hemorrhage and resuscitation swine study. Through this optimization process, we evaluated different time-lengths of prediction windows, machine learning model architectures, and data normalization approaches. Models were successful at predicting Percent Estimated Blood Loss in blind swine subjects with coefficient of determination values exceeding 0.8. This provides evidence that Percent Estimated Blood Loss can be accurately derived from non-invasive signals, improving its utility for trauma care and casualty triage in the pre-hospital and emergency medicine environment.

## 1. Introduction

Hemorrhage is the leading cause of preventable death in both military and civilian trauma care [1,2]. Uncontrolled hemorrhage can lead to hemorrhagic shock, a condition characterized by significant blood loss that impairs oxygen delivery to tissues, forcing cells into anaerobic metabolism. This process generates lactic acid, lowers blood pH, and can lead to widespread cellular necrosis and death if untreated [3]. Hemorrhagic shock is classified into four stages (Class I-IV) based on injury severity defined by volume of blood loss and other physiological symptoms [4]. Estimated blood loss (EBL) is commonly assessed in operating rooms using visual estimation techniques to guide blood transfusion amounts; EBL can be critical when treating hemorrhagic shock but is often either under- or over-estimated [5]. The error when calculating EBL is further compounded in chaotic, high-stress environments where first responders are required to provide initial interventions with minimal resources.

Recent conflicts, illustrated by both the Russo-Ukrainian War and the Israeli–Gaza War, have exposed the evolving nature of combat between near-peer adversaries, where air superiority maybe contested or denied, and civilian infrastructure becomes directly involved. In these high-threat environments, traditional evacuation routes are frequently targeted or inaccessible, severely complicating medical response efforts. Hospitals near conflict zones have faced overwhelming surges, supply shortages, and personnel constraints, forcing frontline responders, both civilian and military, to deliver care under austere conditions with prolonged delays. As a result, casualties often remain at or near the point of injury for extended durations, requiring prolonged field care in settings ranging from remote battlefields to urban centers under fire [6,7]. In these mass casualty scenarios, prompt intervention is crucial for mitigating the effects of hemorrhage. However, diagnostic capabilities are often limited in pre-hospital settings, complicating decision-making. Accurate triage and early diagnosis could not only enhance on-scene management but also streamline the activation of trauma system protocols, such as the Mass Transfusion Protocol (MTP). MTPs involve administering 10 or more units of whole blood or packed red blood cells and requires time for preparation and delivery, with median activation and first-unit delivery times of 8 and 9 min, respectively [8].

Advancing tools that enable rapid and precise detection of critical blood loss triggering timely activation of a MTP could significantly reduce delays in life-saving care, particularly in high-acuity or resource constrained environments across both civilian and military trauma settings. In response, properly developed machine learning (ML) models have the potential to act as an early indicator of the onset of hemorrhagic shock. As a subset of artificial intelligence, ML models have predictive capabilities after gathering learning parameters from a training dataset and using these patterns to make decisions on new, often blind data. Advanced ML algorithms can provide an advanced toolset for data analysis and interpretation of biological signals [9]. ML models have shown promise in detecting and predicting hemorrhagic shock using metrics such as Percent Estimated Blood Loss (PEBL) [10,11]. The PEBL metric quantifies blood loss as a percentage of total blood volume based on weight-derived blood volume constants. While these models outperform traditional metrics in speed and accuracy, their reliance on the blood pressure waveform from an invasive arterial catheter makes them impractical for field use.

Non-invasive diagnostic tools may offer a practical solution for remote field applications where conventional methods are unreliable due to physiological compensatory mechanisms that mask early signs of hemorrhagic shock. These mechanisms temporarily prioritize perfusion to vital organs but are unsustainable, leaving providers without timely indicators of patient status [12]. This underscores the need for a rapid, accurate metric derived from non-invasive sensing that can effectively provide clinical decision-support during triage.

This study explores the application of ML models trained from non-invasive photoplethysmography (PPG) signals to predict PEBL. The PPG signal was chosen as a PPG sensor can be integrated into portable and inexpensive cardiovascular health assessment wearables that can easily be transported and deployed in the pre-hospital environment for early hemorrhage detection [13]. Beyond only tracking hemorrhage, we evaluated if the PPG-based PEBL metric continued to track relative fluid balance during hemorrhagic shock resuscitation. By leveraging advanced feature extraction and model optimization, we aimed to demonstrate the feasibility of this non-invasive approach to enhance triage and intervention in hemorrhagic shock scenarios.

## 2. Materials and Methods

### 2.1. Animal Data Capture

For the purposes of this study, data was pooled from two approved animal research protocols (*n* = 23 subjects total). Research was conducted in compliance with the Animal Welfare Act, the implemented Animal Welfare regulations and the principles of the Guide for the Care and Use of Laboratory Animals. The Institutional Animal Care and Use Committee at the United States Army Institute of Surgical Research approved all research conducted in the protocols. The facility where these research projects were conducted is fully accredited by AAALAC International. All animals throughout both studies were maintained under a surgical plane of anesthesia and administered analgesia, Buprenorphine SR (Wedgewood Pharmacy; Swedesboro, NJ, USA), once pre-procedure.

These preclinical pilot studies were aimed at demonstrating the proof-of-concept for various physiological closed-loop controllers in hemorrhagic shock resuscitation [14,15]. The studies were conducted using a swine (*Sus scrofa domestica*) model, of intact female Yorkshire crossbred swine, approximately 4 months old, weighing around 40 kg. Swine were selected as the large animal model of choice due to its recognition as an appropriate candidate for research in hemorrhage studies due to similarities in physiology to humans, particularly the cardiovascular system [16,17]. Animals were instrumented with central catheters (Arrow International, Morrisville, NC, USA) at the carotid artery and femoral vessels for purposes of tracking pressure, removing fluid during controlled hemorrhage, and delivering fluids during resuscitation. The swine subjects were allowed to stabilize during a baseline period after undergoing a laparotomy to remove the spleen prior to a controlled hemorrhage event to a target mean arterial pressure (MAP) of 35 mmHg. Animals were held at the MAP target for up to a set duration of 90 min or until blood lactate levels reached 4 mmol/L. Next, subjects, for one animal protocol (*n* = 11 subjects), were resuscitated with whole blood to a target MAP value, followed by holding at target MAP with crystalloid for 2 h [14]. For the second protocol (*n* = 12 subjects), animals were resuscitated with whole blood to a target MAP value until 2 units were used, followed by using crystalloid for the duration of a 1 h resuscitation window. Each of these resuscitations used an automated closed loop adaptive resuscitation controller to personalize flow rates and overall fluid delivered [14]. After completion of subsequent study events for each protocol, the swine subjects were humanely euthanized.

### 2.2. Data Processing

PPG waveform data collected from all animal subjects (*n* = 23) along with a cumulative record of blood removed and fluid infused was further analyzed for development of training datasets. For the initial PEBL model development, the datasets used were truncated to a 10 min baseline period, the hemorrhage event, and hypovolemic hold. For model testing purposes, the data captured during the resuscitation phase of each study were included for evaluating if the PEBL models were suitable for tracking resuscitation fluid balance. The PPG signal was recorded at 500 Hz, and the blood loss and fluid infusion data at 0.2 Hz. To harmonize these datasets and enable better alignment, both were resampled to a uniform rate of 100 Hz.

The cumulative blood loss data were then used to calculate the ground truth (GT) PEBL metric values. PEBL was calculated by dividing the hemorrhaged volume at a point in time by the estimated total blood volume of the subject [10,18], giving a value from 0 to 1 representing the percentage of total blood volume lost over time. Blood volume was estimated at 60 mL per kilogram of body weight (40 kg) across all animals [19]. The GT PEBL values were calculated for each subject and paired with the PPG signals to serve as the response variable for the ML model training.

For the fluid resuscitation study phase, two fluid types were used—whole blood and crystalloid. To adjust the PEBL equation to account for overall fluid balance during resuscitation, the reinfused volume started accumulating from the maximum blood loss measured. During the whole blood resuscitation, it was assumed that all infused volume contributed to reducing the PEBL metric. When the resuscitation infusate was swapped from whole blood to crystalloid, the PEBL calculation was adjusted by dividing the infused volume by three, based on the widely used 3:1 crystalloid resuscitation rule [20]. This is due to how crystalloid fluids distribute across the intravascular and extravascular volumes resulting in less fluid remaining in the intravascular space for combatting the hypovolemic effects [21]. When the fluid necessary to reach and maintain target MAP during the resuscitation surpasses the maximum blood loss achieved during the hemorrhage phase, the fluid balance GT would swap signs as the data would switch from a net blood loss (defined as a positive value for the metric) to an overall fluid increase (defined as a negative value for the metric) in the fluid balance calculation.

### 2.3. Feature Engineering

The features used in this study were adapted from Chowdhury et al. [22], with modifications to focus on blood loss prediction rather than blood pressure estimation. A total of 58 features were extracted from the PPG waveform, organized into five primary categories: signal morphology features, PPG width features, derivative-based-features, frequency domain features, and statistical features. 

### 2.4. Model Selection

The selection of ML models was guided by the need to balance model interpretability, computational efficiency, and performance. As such four models architectures were evaluated: Extreme Gradient Boosting (XGB) [23], Random Forest (RF) [24], Elastic Net (ENET) [25], and Support Vector Regression (SVR) [26]. Each of these models offers distinct advantages described below that align with the objectives of this study and the characteristics of the dataset. The process used for down-selection of model architectures and model hyperparameters is diagrammed in Figure 1 and the hyperparameters are described in Table 1.

Both RF and XGB are tree-based models. XGB is a commonly used ML model that operates by creating an ensemble of decision trees in a sequential manner, optimizing a specific loss function at each iteration. A RF model was included due to its ability to handle complex, nonlinear relationships between features. It works by combining the predictions of multiple decision trees, reducing the risk of overfitting while maintaining high accuracy. ENET is a linear regression model that combines L1 and L2 regularization to perform feature selection while maintaining a balance between bias and variance. Its linear nature ensures interpretability, making it suitable for gaining insight into the importance of individual features. SVR models contain kernels which allow the model to map input data to higher-dimensional spaces where linear relationships can be identified. This is beneficial for datasets with nonlinear patterns that may not be captured by a linear model.

### 2.5. Performance Evaluation

The performance of the four chosen ML models—XGB, RF, ENET, and SVR—were analyzed based on their ability to predict PEBL based on features extracted from the PPG signal, which is reflected in their calculated coefficient of determination (R^2^). R^2^ was chosen as it provides a direct measure of the model’s fit, or how well the model explains the relationship between the features and the target variable (PEBL). Each model was tested under various conditions described below and summarized in Figure 1. For each unique training run, data from one animal subject was removed from the training data to be used as a blind test subject. Each model was then trained using the remaining animal data (*n* = 22) and validated against the hold-out test data. For each of these training runs, a R^2^ score, mean squared error (MSE), and mean absolute error (MAE) were calculated by comparing the predicted PEBL values to the GT data. This process was repeated for each animal subject, resulting in 23 unique performance metrics for each.

### 2.6. Model Optimization

The four models were initially tested with various features extracted from sampling windows of the PPG signal of 5, 10, 30, and 60 s segments to provide a varying range of testing scenarios. The features of each respective sampling window were averaged per segment. These four training datasets were then fed into the four respective ML models. The average R^2^, MSE, and MAE of these four models against GT PEBL were calculated. For each model, the optimal sampling window was selected based on average R^2^ scores. Each model architecture was compared based on their optimal sampling window and the top three models were selected for further evaluation.

Next, we assessed the effect of feature normalization on model performance. The features were normalized for the selected models to a 0-to-1 scale for each test subject. These normalized features were used as inputs to the ML models and the R^2^, MSE, and MAE performance metrics were calculated. The higher R^2^ values between non-normalized and normalized features were selected for each model architecture and advanced to test the effects of optimization on the advancing models. Similarly to the previous step, only the top two of the three remaining model architectures were selected for additional evaluation for PEBL predictions.

To evaluate the effects of hyperparameter tuning for each remaining model architecture, each model’s parameters were tuned using a Bayesian optimization process to select the best parameters for each respective model [27]. Hyperparameters tuned for each model are listed in Figure 1, described in Table 1, and additional information can be found on Scikit-learn [28] for RF, ENET, and SVR while additional XGB information can be found at DMLC XGBoost [23]. For each training dataset, four subjects were removed for validation during the Bayesian optimization process. Models were trained on the remaining subjects for a total of 100 training iterations, with different combinations of parameters being evaluated throughout the optimization process. Root mean squared error (RMSE) values were calculated from predictions made on the validation data, and the best performing model parameters based on RMSE were selected. Models were then trained with these selected hyperparameters to compare overall model performance against the non-tuned model versions. The effects of hyperparameter tuning on each model were compared against the untuned model version and one was selected for each model architecture based on R^2^ scores. Lastly, a single model was selected based on R^2^ values after the downselection of the different sampling windows, non-normalized versus normalized features, and finally non-optimized versus optimized hyperparameters.

### 2.7. Evaluation of Model Tracking Fluid Balance During Resuscitation

To evaluate utility beyond the isolated hemorrhage phases, the best-performing models based on the above optimization for each ML architecture type were further tested using extended experimental data, which included both the hemorrhage phase that was the focus of model training and a resuscitation phase which was not included in training datasets. While the above optimization downselected to a single model architecture, all four model types were used in this phase of the study to determine if any model types were better suited for generalized performance. For the downselected model architectures, setup was based on optimization results prior to their downselection, with non-normalized, no hyperparameter tuning being default options if the model did not advance to these optimization steps. Since this broader testing context includes fluid return (i.e., resuscitation), the outcome variable was redefined from PEBL to Fluid Balance, representing the net volume status of the subject over time. The MAE, MSE, and R^2^ were calculated for all the models used in this testing phase to allow performance evaluation between all the tested models.

### 2.8. Statistical Analysis of Model Performance

The primary performance metric for this study was R^2^, so statistical analyses were performed to assess significant differences between each model type and model configuration. For each model training phase, 23 replicates were analyzed corresponding to 23 blind swine subjects. Shapiro–Wilk test was used to evaluate if these datasets were normally distributed, and results were consistently non-normally distributed. As such, when more than two experimental groups were being compared, Friedman test with post hoc Dunn’s multiple comparison test was used, wherein *p*-values of less than 0.05 indicated significant differences between R^2^ values for the compared groups. When comparing only two groups, Wilcoxon matched pairs ranked test was used where a *p*-value less than 0.05 indicated significant differences between compared R^2^ values. When applicable, figure captions describe statistical tests performed and statistical significances are denoted.

## 3. Results

### 3.1. Sampling Window Optimization

The XGB, RF, ENET, and SVR model were trained and tested on blind subjects using sampling windows of 5, 10, 30, and 60 s. The average R^2^ values across each time sampling window was 0.799 for 5 s, 0.789 for 10 s, 0.773 for 30 s, and 0.789 for 60 s (Figure 2). However, differences were slight within each model architecture, with only SVR 5 s being significantly higher than SVR 60 s (Figure 2D). On average, the features extracted from the 5 s sampling window had the best performance across the different models, except for ENET where the 60 s sampling window performed the best. Comparing the best sampling window for each configuration, XGB (5 s) and RF (5 s) R^2^ metrics were significantly better than both ENET (60 s) and SVR (5 s) (Figure 2E). Differences between XGB (0.839) and RF (0.846) were not significantly different. Tabular results for MAE, MSE, and R^2^ are shown in Appendix A.

Overall, ENET was the worst performing model, based on R^2^ scores, and was removed from consideration prior to the following stages of optimization. The 5 s sampling window was the optimal configuration for the remaining models which were used for the next phase, incorporating normalization. Representative subjects demonstrating the results of ENET with the four different sampling windows are shown in Figure 3 for visualization of the results. Similar representations for the RF, XGB, and SVR model architecture are shown in Appendix A.

### 3.2. Data Normalization

After downselecting from 4 to 3 ML model architectures, the effects on model performance using normalized versus non-normalized features were tested. The average R^2^ of the normalized features across all models was 0.778 while the average R^2^ across all models using non-normalized features was 0.823 (Figure 4). On average, the non-normalized features had the highest performance; however, only RF R^2^ differences were significantly different with non-normalized outperforming normalized configurations (Figure 4B). Within the non-normalized features, XGB and RF significantly outperformed SVR with R^2^ values of 0.839, 0.846, and 0.783, respectively (Figure 4D). Summary results for each model architecture’s performance between normalization and non-normalization are shown in Table 2.

The non-normalized configuration was selected as the variable that continued into the next stage of optimization. SVR, as the worst performing model, was not considered in the following stages of optimization. Representative subjects demonstrating the PEBL predictions with SVR for both normalization and non-normalization parameters are shown in Figure 5. Similar representations for the RF and XGB model architecture are shown in Appendix A.

### 3.3. Hyperparameter Tuning

Using the downselected models and the best performing variables—5 s sampling windows and non-normalized features—the ML models, XGB and RF, were optimized using Bayesian optimization to tune the hyperparameters in order to see the effects on model performance. The average R^2^ of the optimized model architectures was 0.871 while the average R^2^ of the non-optimized was 0.843 (Figure 6). The XGB model was significantly improved through the optimization process (Figure 6A) while the RF models were not significantly different. Within the optimized variable, the XGB ML model had the strongest R^2^ with an R^2^ of 0.872, only slightly improved over RF at 0.870 (Figure 6C). These differences were not statistically significant. Summary results for the model architecture’s performance between optimization and non-optimization are shown in Table 3 below.

The optimized XGB model was the model that was downselected across all variables to be the highest performing model based on R^2^. Representative subjects demonstrating the results of the RF comparing optimization versus non-optimization are shown in Figure 7A,B for visualization of the results. Additionally, representative PEBL predictions for two subjects of the best, overall performing model—the XGB model with a 5 s sampling window, non-normalized features, and optimized hyperparameters—are shown in Figure 8A,B.

### 3.4. Extended Model Performance

Lastly, we evaluated the ability of the PEBL models, trained using their best-performing configurations, to predict fluid balance during the resuscitation phase of the animal study. From representative training curves for each model type it was evident that performance continued to improve as more training data was provided but the rates of improvement were not consistent across each model (Appendix A). Although fluid balance was not a primary outcome and models were not explicitly trained for this target, they still demonstrated moderate predictive capacity (Figure 9). Among the optimized models, RF (5 s, non-normalized, tuned) achieved the highest R^2^ at 0.614, followed closely by XGB (5 s, non-normalized, tuned) with an R^2^ of 0.609. SVR (5 s, non-normalized) and ENET (60 s, non-normalized) performed less strongly, with R^2^ values of 0.413 and 0.419, respectively (Table 4). While R^2^ performance for RF and XGB were not significantly different, both were significantly stronger than SVR, and RF performance was significantly higher compared to ENET (Figure 10). These results suggest that while model performance was lower than for the hemorrhage region, the optimized architectures retained sufficient generalizability to support meaningful predictions in the resuscitation phase of the study.

## 4. Discussion

In both military and civilian trauma care, prompt reductions in ongoing blood loss and surgical intervention remain the most effective treatments for hemorrhagic shock. Delays in initiating these treatment modalities are associated with worse clinical outcomes [29], thus demonstrating the need for early and accurate detection of impending hemorrhagic shock. This is difficult to assess with traditional vital signs, as physiological compensation mechanisms can mask ongoing bleeding until treatment is no longer effective and the clinical outcome is worsened. Previous work developed a metric that accurately predicted blood loss during hemorrhage in a large animal study. This metric, PEBL, was initially developed using arterial waveform data, which requires an invasive arterial line catheter. The study expanded upon that work by creating and testing ML models capable of predicting PEBL using non-invasive PPG sensors.

Testing the various sampling window lengths demonstrated that smaller windows of time were more effective at capturing vital features and allowed the ML models to have improved regression performance. The features captured using the 5 s sampling windows had the highest R^2^ value across all the tested models. The worst performing model, ENET, failed to go through further optimization, likely due to having less flexibility due to its linear assumptions when compared to the other 3 model architectures [30]. After taking the 5 s sampling windows onto the next stage, models were fed with normalized and non-normalized features. These two parameters had significant differences in results, with the non-normalized features being superior with an R^2^ of 0.823. This occurred across all the model architectures in this step. The SVR model performed the worst out of the group of models tested, likely due to being susceptible to scaling as SVRs rely on kernel-based distance metrics [31].

Optimization had a significant impact on model performance for both RF and XGB architectures with a 5 s sampling window and non-normalized features. The improvement from baseline to optimized models reinforces the importance of hyperparameter tuning in achieving optimal predictive accuracy. The outcome of the XGB architecture being the most benefitted highlights that while optimization is beneficial, its effectiveness is model-dependent and should be carefully applied based on each model’s architecture and response to feature variability. These results underscore the effectiveness of ensemble methods in modeling dynamic physiological data and highlight their suitability for real-time or near-real-time applications.

The ability to accurately predict PEBL from PPG waveforms has important clinical implications, particularly in the early detection and management of hemorrhage and hemorrhagic shock. Timely and accurate estimation of blood loss is a critical component of trauma care, as delays in diagnosis can result in irreversible shock or death. The findings of this study demonstrate that ML models, particularly RF and XGB, can provide reliable, accurate predictions of blood loss from non-invasive PPG signals. The superior performance of XGB as well as RF models can be attributed to their ability to capture nonlinear relationships and interactions among data, including PPG-derived features [23,32]. This represents a meaningful advancement toward scalable, field-deployable solutions for real-time monitoring in high-stress, resource-limited environments such as emergency transport, operating rooms, and battlefield conditions.

Although the highest performance was observed in the 5 s sampling windows, the performance of models in the 30 and 60 s segments suggests that longer feature windows may still offer value in certain clinical contexts. Capturing sustained physiological patterns over time could improve detection of subtle trends during resuscitation or recovery phases, where rapid fluctuations are less pronounced. These longer windows may serve as a useful complement to short-window predictions, especially in continuous monitoring settings where stability and signal smoothing are desirable.

To assess the applicability of our models beyond hemorrhage-only scenarios, we conducted a secondary evaluation using data from the full experimental timeline—encompassing both hemorrhage and active resuscitation. In this broader context, the target variable was redefined as Fluid Balance, which captures the net fluid state of the subject, integrating both blood loss and fluid replacement over time. Due to the GT PEBL making certain assumptions in the resuscitation region, like the infusion volume being 1:1 when the infusate is whole blood and 3:1 when using crystalloid [20], as well as the fixed blood volume assumption per subject weight, the GT may be skewed due to error being introduced. These assumptions, however, stemmed from research that reflects real-world trauma progression and treatment.

Results from this expanded analysis confirmed the robustness of the top-performing models that advanced to the last round of optimization. The RF and XGB models, particularly in the optimized 5 s configuration using non-normalized features, continued to deliver the highest R^2^ values, 0.614 and 0.609, respectively. The retention of predictive strength across both phases indicates that features derived from short PPG windows capture physiologically relevant signal patterns tied not just to loss but to overall volume shifts, a critical insight for dynamic monitoring environments like medical evacuations or emergency departments.

There are some limitations with the currently developed non-invasive blood loss tracking metrics. First, they are built using limited datasets from animal studies, specifically under a splenectomy hemorrhage model. As a result, the models may be biased toward those specific experimental conditions. These datasets are small currently and more generalized prediction models may be possible with larger datasets; this is evidenced by learning curves during training wherein performance continued to improve as more data was presented. Broader validation will require diverse datasets, including other types of hemorrhage and eventual translation to human subjects. Second, while the PEBL metric demonstrates a clear correlation with hemorrhage progression, its clinical utility remains to be validated. Future animal studies with well-defined outcome measures or access to human datasets with tracked patient outcomes will be essential to confirm whether PEBL can non-invasively track with real-world hemorrhage severity and casualty prognosis.

The next step for this work is to evaluate the viability of these models in real-time triage tracking. Future investigations will explore whether these metrics can also support resuscitation decision-making, either in initial treatment or during extended care. Ongoing animal studies will examine longer-term recovery and outcome metrics to determine how well PEBL reflects the full trajectory of hemorrhage and resuscitation. Additionally, future training will incorporate expanded datasets to further generalize the model’s performance across a wider range of bleeding conditions.

## 5. Conclusions

In summary, this proof-of-concept study provides strong evidence that PEBL can be accurately predicted using non-invasive sensors with ML which may have significant implications for improving triage decision-making in pre-hospital and acute care environments. This finding reinforces the versatility of the proposed models and their potential utility for guiding hemorrhage resuscitation, where real-time fluid status monitoring could enhance clinical decisions during and after hemorrhage. It also opens the door for future studies aimed at modeling fluid responsiveness, over-resuscitation risk, or shock recovery trajectories, using similar non-invasive sensing and ML-based prediction techniques which will be explored in future studies.

## Figures and Tables

**Figure 1 bioengineering-12-00833-f001:**
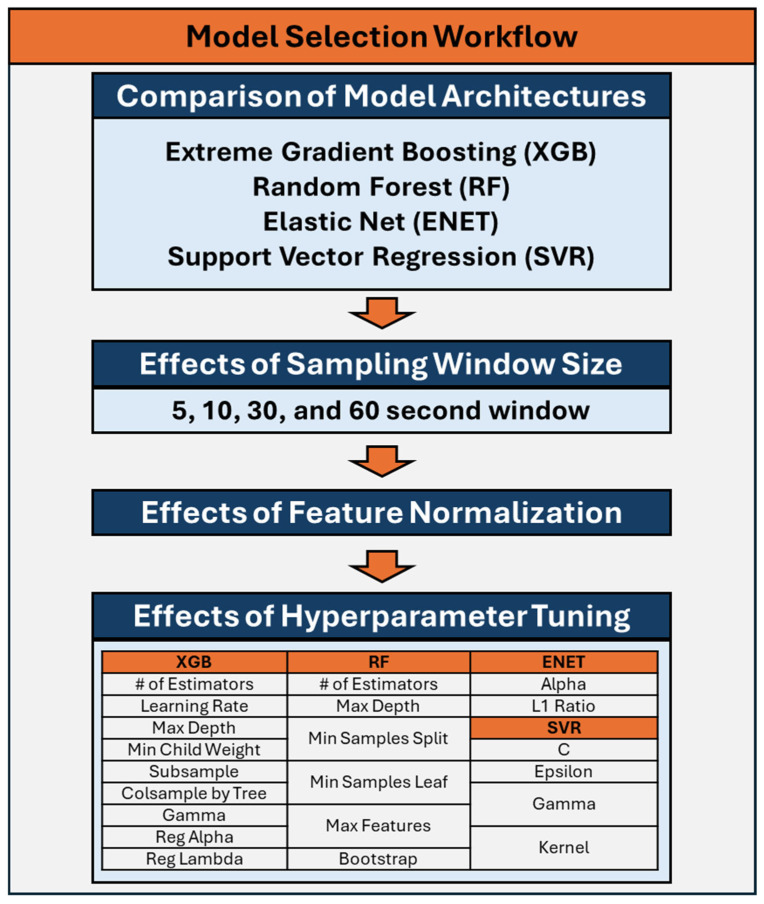
Workflow for down-selection and optimization of the machine learning models for prediction of Percent Estimated Blood Loss.

**Figure 2 bioengineering-12-00833-f002:**
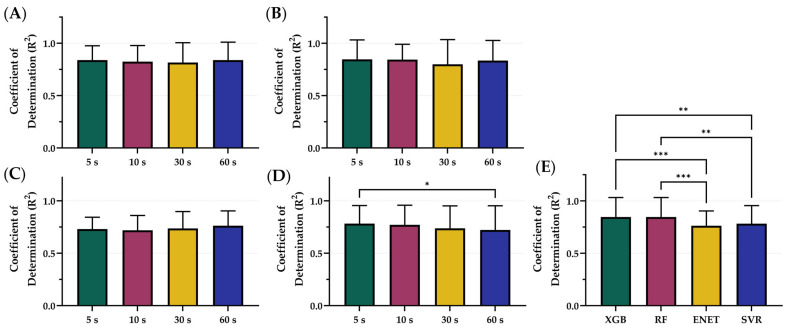
Comparison of Coefficient of Determination (R^2^) scores for different sampling window configurations. Four sampling windows—5, 10, 30, and 60 s—were compared for (**A**) XGB, (**B**) RF, (**C**) ENET, and (**D**) SVR. (**E**) Comparison of optimal model configurations based on sampling window. Results are shown as mean values (*n* = 23 replicates) with error bars denoting standard deviation. Statistical differences are shown when present (*p*-value less than 0.05 [*], 0.01 [**], and 0.001 [***]) based on Friedman test, post hoc Dunn’s multiple comparison tests.

**Figure 3 bioengineering-12-00833-f003:**
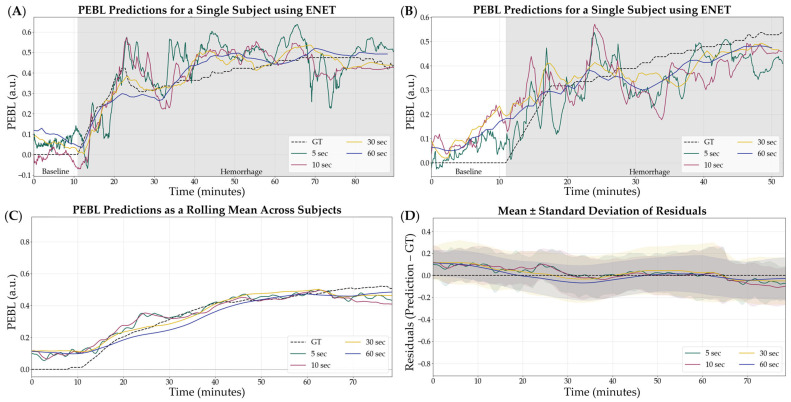
Effect of sample window size on ENET model performance. Four different sampling window sizes were evaluated—5, 10, 30, and 60 s—compared against calculated GT estimates. (**A**,**B**) Representative single subject results for PEBL vs. time during the baseline and hemorrhage scenarios. (**C**) Average results for ENET for all swine subjects (*n* = 23) for each sample window size and ground truth. (**D**) Average residuals across all subjects (*n* = 23) for ENET models, with shaded regions denoting standard deviation, black dashed line represents zero, a perfect prediction.

**Figure 4 bioengineering-12-00833-f004:**
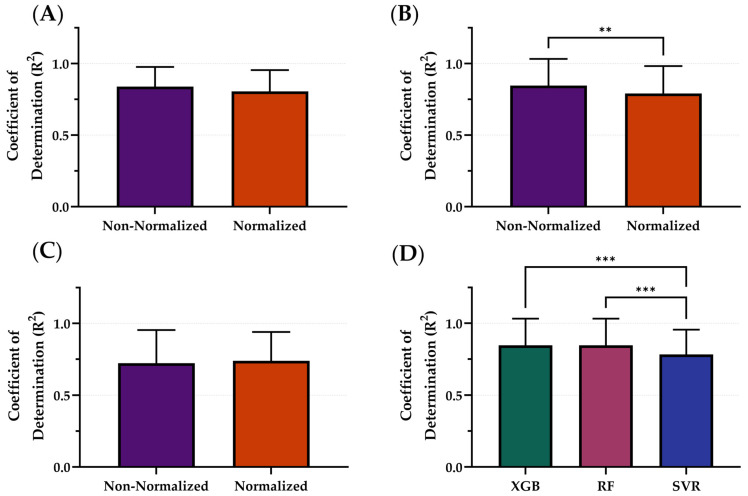
Comparison of Coefficient of Determination (R^2^) scores with and without normalization for different model types. The effects of normalization for (**A**) XGB, (**B**) RF, and (**C**) SVR. (**D**) Comparison of optimal model configurations for each. Results are shown as mean values (*n* = 23 replicates) with error bars denoting standard deviation. Statistical differences are shown when present (*p*-value less than 0.01 [**] and 0.001 [***]) based on (**A**–**C**) Wilcoxon matched-pairs ranked test or (**D**) Friedman test, post hoc Dunn’s multiple comparison tests.

**Figure 5 bioengineering-12-00833-f005:**
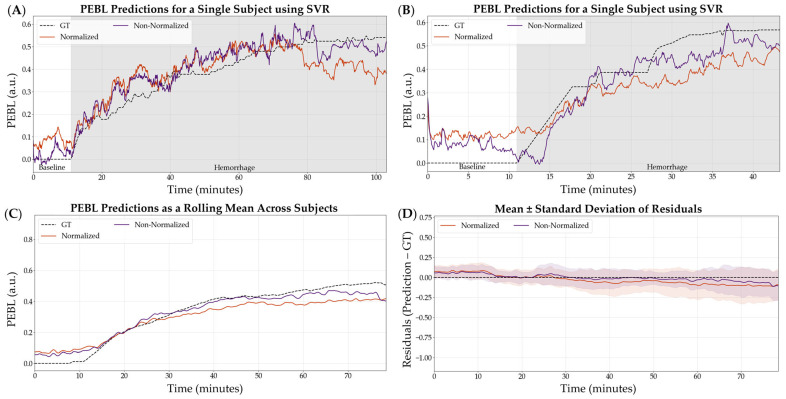
Effect of normalization vs. non-normalization on the SVR model performance using a 5 s feature window. (**A**,**B**) Representative single subject results for PEBL vs. time during baseline and hemorrhage states for normalized and non-normalized configurations with the ground truth. (**C**) Average results for SVR for all swine subjects (*n* = 23) for normalized and non-normalized configurations with the ground truth. (**D**) Average residuals across all subjects (*n* = 23) for SVR models, with shaded regions denoting standard deviation and black dashed line at zero representing a perfect prediction.

**Figure 6 bioengineering-12-00833-f006:**
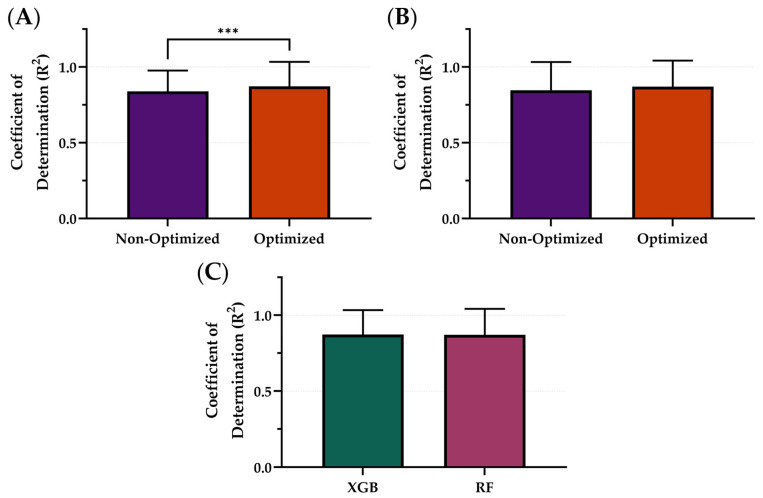
Comparison of Coefficient of Determination (R^2^) scores with and without optimization for different model types. The effects of optimization for (**A**) XGB and (**B**) RF. (**C**) Comparison of optimal model configurations for each. Results are shown as mean values (*n* = 23 replicates) with error bars denoting standard deviation. Statistical differences are shown when present (*p*-value less than 0.001 [***]) based on Wilcoxon matched-pairs ranked test.

**Figure 7 bioengineering-12-00833-f007:**
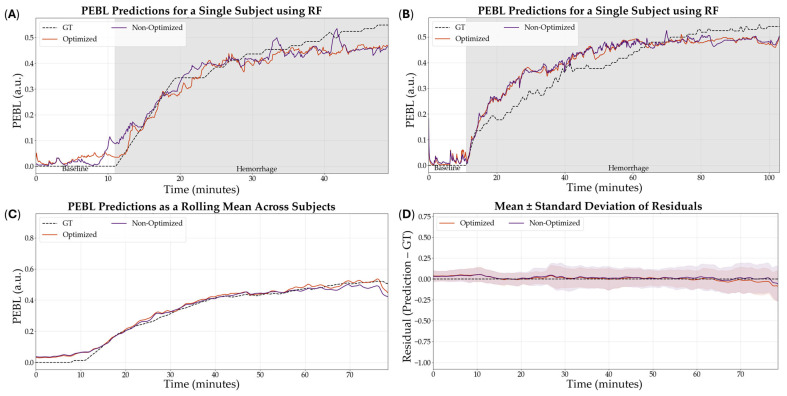
Effect of hyperparameter tuning on RF model performance. (**A**,**B**) Representative single subject results for PEBL vs. time during baseline and hemorrhage scenarios for the Bayesian optimized and non-optimized configurations along with the ground truth. (**C**) Average results for RF for all swine subjects (*n* = 23) for the Bayesian optimized and non-optimized configurations. (**D**) Average residuals across all subjects (*n* = 23) for RF models, with shaded regions denoting standard deviation, the black dashed line at zero represents perfect predictions.

**Figure 8 bioengineering-12-00833-f008:**
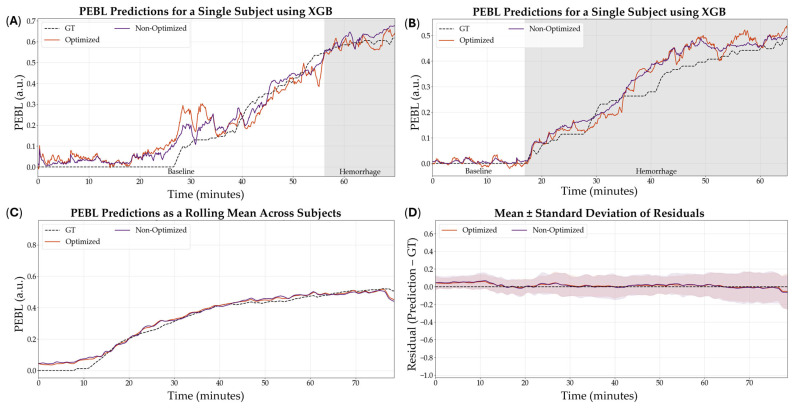
Effect of hyperparameter tuning on the XGB model performance. (**A**,**B**) Representative single subject results for PEBL vs. time during baseline and hemorrhage scenarios for the Bayesian optimized and non-optimized configurations with the respective ground truth. (**C**) Average results for XGB for all swine subjects (*n* = 23) for the Bayesian optimized and non-optimized configurations. (**D**) Average residuals across all subjects (*n* = 23) for XGB models, with shaded regions denoting standard deviation, the black dashed line at zero represents perfect predictions.

**Figure 9 bioengineering-12-00833-f009:**
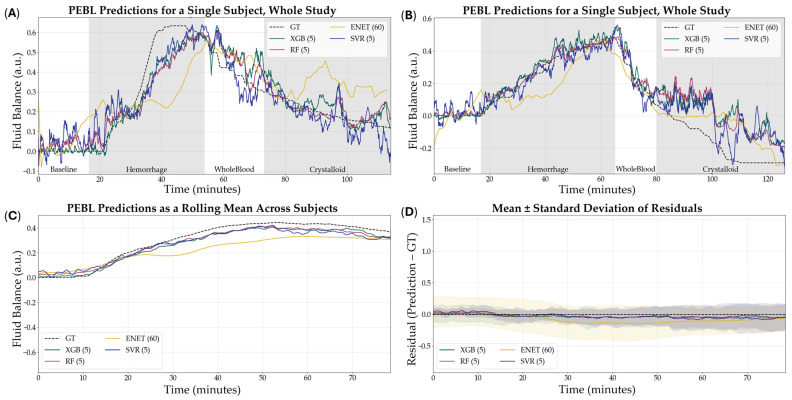
Results of all models to predict fluid balance during hemorrhage and resuscitation. (**A**,**B**) Representative single subject results for PEBL vs. time during the baseline, hemorrhage, and resuscitation staged with the respective ground truth. (**C**) Average results for all models for all swine subjects (*n* = 23) across the entire study. (**D**) Average residuals across all subjects (*n* = 23) for all models, with shaded regions denoting standard deviation and the black dashed line at zero represents perfect predictions.

**Figure 10 bioengineering-12-00833-f010:**
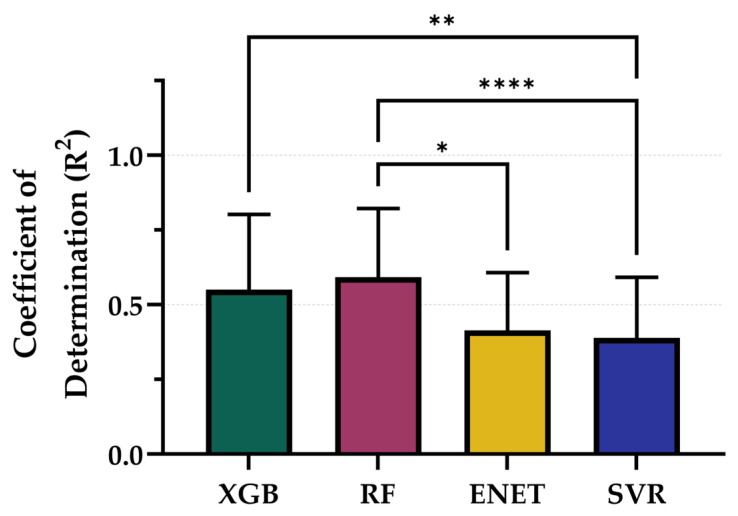
Comparison of Coefficient of Determination (R^2^) scores for each model across the entire study paradigm. Results are shown as mean values (*n* = 23 replicates) with error bars denoting standard deviation. Statistical differences are shown when present (*p*-value less than 0.05 [*], 0.01 [**], and 0.0001 [****]) based on Friedman test, post hoc Dunn’s multiple comparison tests.

**Table 1 bioengineering-12-00833-t001:** Overview of hyperparameters for machine learning models used for the prediction of Percent Estimated Blood Loss for the four model architectures: Extreme Gradient Boosting (XGB), Random Forest (RF) [24], Elastic Net (ENET), and Support Vector Regression (SVR).

Parameter	Definition	XGB	RF	ENET	SVR
# of Estimators	Number of trees or models to train in the ensemble	✔	✔		
Learning Rate	How much each tree contributes to the final prediction	✔			
Max Depth	The maximum depth of each decision tree	✔	✔		
Min Child Weight	Minimum sum of instance weights needed in a child node to allow a split	✔			
Subsample	Fraction of training data randomly sampled for each tree to prevent overfitting	✔			
Colsample by Tree	Fraction of features randomly sampled for each tree	✔			
Gamma	Regularization parameter; in XGB, it sets the minimum loss reduction to make a split; in SVR, it defines how far the influence of a single training example reaches	✔			✔
Reg Alpha	L1 regularization term on weights to encourage sparsity	✔			
Reg Lambda	L2 regularization term on weights to reduce complexity	✔			
Min Samples Split	Minimum number of samples required to split an internal node		✔		
Min Samples Leaf	Minimum number of samples required to be at a leaf node		✔		
Max Features	Number of features to consider when looking for the best split		✔		
Bootstrap	Whether bootstrap samples are used when building trees		✔		
Alpha	Overall strength of regularization (combines L1 and L2)			✔	
L1 Ratio	Ratio of L1 and L2 regularization			✔	
C	Regularization parameter that balances margin maximization and error				✔
Epsilon	Tolerance within which no penalty is given in the loss function for errors				✔
Kernel	Specifies the kernel type to be used (e.g., linear, RBF)				✔

**Table 2 bioengineering-12-00833-t002:** Summary normalization model performance results. Results for 3 model architectures are shown—RF, XGB, and SVR. Average results across all subjects are shown for MAE, MSE, and R^2^. Bold R^2^ values represent the top performing configuration for each model while the green and red shading indicate the models that were selected or removed from the next optimization steps, respectively.

5 s Sampling Window						
	Normalized			Non-Normalized		
Model	MAE	MSE	R^2^	MAE	MSE	R^2^
XGB	0.100 ± 0.056	0.0190 ± 0.022	0.805 ± 0.150	0.0909 ± 0.059	0.0171 ± 0.026	**0.839 ± 0.138**
RF	0.0988 ± 0.072	0.0228 ± 0.034	0.791 ± 0.191	0.0845 ± 0.063	0.0173 ± 0.027	**0.846 ± 0.186**
SVR	0.113 ± 0.072	0.0245 ± 0.040	0.739 ± 0.202	0.0908 ± 0.052	0.0170 ± 0.024	**0.783 ± 0.173**
Average	0.104 ± 0.067	0.0221 ± 0.032	0.778 ± 0.181	0.0887 ± 0.058	0.0171 ± 0.026	0.823 ± 0.166

XGB = Extreme Gradient Boosting; RF = Random Forest; SVR = Support Vector Regression; MAE = Mean Absolute Error; MSE = Mean Squared Error; R^2^ = Coefficient of Determination.

**Table 3 bioengineering-12-00833-t003:** Summary results for hyperparameter tuning on model performance. The final two model architectures are shown—RF and XGB. Average results across all subjects are shown for MAE, MSE, and R^2^. Bold R^2^ values represent the top performing configuration for each model while the green and red shading indicate the models that were selected or removed from the next optimization steps, respectively.

5 s Sampling Window, Non-Normalized						
	Optimized			Non-Optimized		
Model	MAE	MSE	R^2^	MAE	MSE	R^2^
XGB	0.0814 ± 0.060	0.0148 ± 0.025	**0.872 ± 0.161**	0.0909 ± 0.059	0.0171 ± 0.026	0.839 ± 0.138
RF	0.0731 ± 0.055	0.0132 ± 0.024	**0.870 ± 0.172**	0.0845 ± 0.063	0.0173 ± 0.027	0.846 ± 0.186
Average	0.0773 ± 0.058	0.0140 ± 0.025	0.871 ± 0.167	0.0877 ± 0.061	0.0172 ± 0.027	0.843 ± 0.162

XGB = Extreme Gradient Boosting; RF = Random Forest; MAE = Mean Absolute Error; MSE = Mean Squared Error; R^2^ = Coefficient of Determination.

**Table 4 bioengineering-12-00833-t004:** Summary results with inclusion of resuscitation on model performance. The best combination of parameters is shown across four model architectures—XGB, RF, ENET, and SVR. Average results across all subjects are shown for MAE, MSE, and R^2^.

	Performance Metrics Across Hemorrhage and Resuscitation Phases		
Model (Setup Parameters)	MAE	MSE	R^2^
XGB (5 s, Non-Normalized, Tuned)	0.147 ± 0.069	0.0429 ± 0.038	0.609 ± 0.253
RF (5 s, Non-Normalized, Tuned)	0.137 ± 0.070	0.0397 ± 0.039	0.614 ± 0.230
ENET (60 s, Non-Normalized, Not Tuned)	0.207 ± 0.213	0.0881 ± 0.265	0.419 ± 0.203
SVR (5 s, Non-Normalized, Not Tuned)	0.166 ± 0.195	0.0504 ± 0.215	0.413 ± 0.194

XGB = Extreme Gradient Boosting; RF = Random Forest; ENET = Elastic Net; SVG = Support Vector Regression; s = seconds; MAE = Mean Absolute Error; MSE = Mean Squared Error; R^2^ = Coefficient of Determination.

## Data Availability

The data presented in this study are not publicly available because they have been collected and maintained in a government-controlled database located at the U.S. Army Institute of Surgical Research. This data can be made available through the development of a Cooperative Research and Development Agreement (CRADA) with the corresponding author.

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
