# Peer review of "Enhancing Trauma Care: Machine Learning-Based Photoplethysmography Analysis for Estimating Blood Volume During Hemorrhage and Resuscitation"

_bioengineering, 2025, doi:10.3390/bioengineering12080833_

Round 1
Reviewer 1 Report
Comments and Suggestions for Authors
-
The models were trained and tested on the same population (n=23 swine) using cross validation. Without independent test dataset, the model is not applicable for generalizability.
-
Testing on the same dataset with unseen subjects is fine, but it’s insufficient to claim robustness.
-
Include statistical comparison (e.g., paired t-tests or Wilcoxon signed-rank tests) between models to justify performance claims.
-
The PEBL ground truth estimation during resuscitation has ambiguity due to fixed blood volume assumptions and fluid correct factors, which may cause systematic bias.
-
Compare the min-max normalization approach with other scaling techniques and justify the selection.
-
What is the regularization penalties of the models like RF or XGB to mitigate the risk of overfitting. Since the subjects (23) are less with high dimensional feature space (58).
-
Include learning curver or bias variance analysis.
-
The paper suffers from verbose and repetitive statements, particularly in the introduction and methods.
-
Many figures (e.g., Figures 2–6) are overloaded with panels (A, B, C, D) without clear labeling or legends. It's difficult to interpret what each subplot specifically refers to. Also, it lacks axis labels and visually unclear.
-
Several references are cited multiple times in the same paragraph (e.g., XGB, RF models) with little added value.
Author Response
1. The models were trained and tested on the same population (n=23 swine) using cross validation. Without independent test dataset, the model is not applicable for generalizability. Thank you for your comment, we agree that additional datasets are required for future development. The limitations and need for additional datasets are described in the discussion. 2. Testing on the same dataset with unseen subjects is fine, but it’s insufficient to claim robustness. Thank you for your comment, evaluating the models using the same dataset does limit generalizability, but the LOSO approach chosen does allow our models to make a completely blind test and the replicates demonstrate that the models can predict accurately even with inter-subject variability. External validation on alternative datasets would provide a stronger claim of robustness but given the data quantity and availability of hemorrhagic datasets, our approach demonstrates robustness better than random data split approaches. 3. Include statistical comparison (e.g., paired t-tests or Wilcoxon signed-rank tests) between models to justify performance claims. The study is more proof-of-concept in nature as the dataset is small as the reviewer mentions. As this study was evaluating a number of factors and making optimization decisions based on goodness-of-fit values which were a non-parametric variable we did not elect to conduct statistical analyses for this study. As that was not the intent of the study, doing that in retrospect would be impractical to the choices made in the study. More statistical metrics have been added to the manuscript in the form of standard deviation values for each table. Due to the number of standard deviation values for the sample window size, we have converted the main result into a graphical figure and the table moved to supplement. We hope this better explains the approach and addresses the reviewer’s concerns 4. The PEBL ground truth estimation during resuscitation has ambiguity due to fixed blood volume assumptions and fluid correct factors, which may cause systematic bias. Thank you for your comment, we agree – we have mentioned how these assumptions will skew the ground truth PEBL calculation as it will introduce error in the discussion. 5. Compare the min-max normalization approach with other scaling techniques and justify the selection. Min-max scaling was selected as the normalization method in this study due to its ability to preserve the relative shape and distribution of features within a fixed range, typically [0, 1]. This characteristic is particularly important for models relying on time-series physiological data, such as photoplethysmography (PPG), where the amplitude and waveform morphology encode key physiological signals. Unlike standardization (z-score normalization), which transforms data based on the mean and standard deviation, min-max scaling maintains the original distribution and dynamic range of the signal features. This ensures that all input features contribute comparably during model training without distorting their relative magnitude — a crucial factor for distance-based and tree-based models like Support Vector Regression (SVR) and Extreme Gradient Boosting (XGB). Furthermore, min-max normalization is well-suited for non-Gaussian or bounded datasets such as those derived from biological signals, where outliers and nonlinear trends may cause z-score normalization to produce misleading feature scales. Since our models were evaluated across multiple animals with varying baseline physiology, min-max scaling offered a consistent way to normalize intra-subject variability while preserving inter-feature relationships critical to prediction accuracy. 6. What is the regularization penalties of the models like RF or XGB to mitigate the risk of overfitting. Since the subjects (23) are less with high dimensional feature space (58). To mitigate the risk of overfitting given the high-dimensional feature space (58 features) and relatively small number of subjects (n = 23), we incorporated multiple forms of regularization and generalization control in our model development. XGB models include both L1 (reg_alpha) and L2 (reg_lambda) penalties, which were tuned via Bayesian optimization to constrain model complexity and promote feature sparsity. Similarly, RF models reduce overfitting through ensembling, bootstrapped data sampling, and random feature selection (max_features) during tree construction. We further applied leave-one-subject-out cross-validation across all 23 subjects to validate model generalizability on blind test data. The consistent performance across subjects and low residual errors suggest that these regularization strategies were effective despite the limited sample size. 7. Include learning curver or bias variance analysis. Thank you for the suggestion, we have included learning curves in the supplement Fig S6 for the top performing configurations for each model type 8. The paper suffers from verbose and repetitive statements, particularly in the introduction and methods. Verbose and repetitive statements have been removed when possible 9. Many figures (e.g., Figures 2–6) are overloaded with panels (A, B, C, D) without clear labeling or legends. It's difficult to interpret what each subplot specifically refers to. Also, it lacks axis labels and visually unclear. We appreciate the feedback and have refined axis labels and figure captions to enhance figure clarity. We believe the panels are complimentary and provide added value by showing representative plots alongside an average plot for all subjects and the prediction residuals. 10. Several references are cited multiple times in the same paragraph (e.g., XGB, RF models) with little added value. Thank you for feedback, we have reviewed the citations that were included. Some of the citations have been maintained as is for clarity and flow in the discussion of each respective model. We will respectfully defer to the editor regarding any further adjustments regarding the citation choices in this manuscript.Reviewer 2 Report
Comments and Suggestions for Authors
An interesting study is devoted to the development of a machine learning model for predicting the estimated percentage of blood loss using a photoplethysmography signal. The models were developed based on data obtained during the study of bleeding and resuscitation in pigs. The data obtained can be of great practical importance and can be further used in the provision of trauma care and triage of victims at the pre-hospital stage and in emergency medical care. There are the following suggestions for this work. 1) the introduction should indicate the percentage of people in whom an incorrect prognostic assessment of the level of blood loss leads (may lead) to adverse consequences. It should also be noted how much the proportion of such people in the field is increasing. 2) in the note to tables 1-5, transcriptions of the abbreviations indicated in the table should be given. 3) in the discussion of the work, it should be discussed in detail to what extent the data obtained on the pig model are comparable to humans. 4) Can there be any gender-specific and age-specific features of using these models?
Author Response
An interesting study is devoted to the development of a machine learning model for predicting the estimated percentage of blood loss using a photoplethysmography signal. The models were developed based on data obtained during the study of bleeding and resuscitation in pigs. The data obtained can be of great practical importance and can be further used in the provision of trauma care and triage of victims at the pre-hospital stage and in emergency medical care. There are the following suggestions for this work.
- the introduction should indicate the percentage of people in whom an incorrect prognostic assessment of the level of blood loss leads (may lead) to adverse consequences. It should also be noted how much the proportion of such people in the field is increasing.
Thank you for your comment. Added verbiage to the introduction regarding the current Russo-Ukrainian and Israeli-Gaza Wars. The lessons learned regarding limitation in triage and treatment due to loss of air superiority or direct attacks to the medical facilities leading to prolonged field care. Due to these limitations, it is important for early detection of blood loss.
- in the note to tables 1-5, transcriptions of the abbreviations indicated in the table should be given.
Agreed, we have added meaning of all abbreviations used in the tables as a footnote or in the captions for added clarity.
- in the discussion of the work, it should be discussed in detail to what extent the data obtained on the pig model are comparable to humans.
Thank you for your comment, we have added a description of why the swine model was chosen due to cardiovascular similarities to humans in the animal section. Additionally, the discussion has content describing the limitations of this type of study and the need for validation with human data.
4) Can there be any gender-specific and age-specific features of using these models?
Thank you for your question, other works in the field with a more variable population have included gender and age specific features. This was outside the scope of our study, and thus the swine population of this study minimized variability (Age, weight, gender, etc.).
Reviewer 3 Report
Comments and Suggestions for Authors
The article is a valuable supplementary material to the main research on the innovative application of machine learning (MO) for the analysis of photoplethysmography (PPG) in predicting blood loss and the need for resuscitation. The work is relevant for military field medicine and emergency medicine, offering potentially fast and non-invasive diagnostic methods.
Questions:
1. Given that previous models for predicting PEBL relied on invasive arterial catheters, why did the authors choose photoplethysmography (PPG) signal as a non-invasive alternative? What specific advantages of the PPG signal make it suitable for field deployment in trauma settings, according to the authors' rationale in the introduction? 2. The models were initially trained on blood loss data, but then tested in the intensive care phase, where the target variable was redefined as "Fluid balance". Why was the performance of the models (R2 about 0.6 for the best models) lower during the resuscitation phase compared to the blood loss phase (R2 > 0.8)? How do the authors explain this decrease in productivity, especially taking into account the assumptions made when calculating the true values of fluid balance (for example, 1:1 for whole blood, 3:1 for crystalloids)?
Author Response
The article is a valuable supplementary material to the main research on the innovative application of machine learning (MO) for the analysis of photoplethysmography (PPG) in predicting blood loss and the need for resuscitation. The work is relevant for military field medicine and emergency medicine, offering potentially fast and non-invasive diagnostic methods.
Questions:
- Given that previous models for predicting PEBL relied on invasive arterial catheters, why did the authors choose photoplethysmography (PPG) signal as a non-invasive alternative? What specific advantages of the PPG signal make it suitable for field deployment in trauma settings, according to the authors' rationale in the introduction?
Thank you for your question, we have added additional details regarding the PPG sensor being able to be integrated into portable form factors, such as wearables, that allow easier transportation and deployment in the field compared to other cardiovascular monitoring systems.
- The models were initially trained on blood loss data, but then tested in the intensive care phase, where the target variable was redefined as "Fluid balance". Why was the performance of the models (R2 about 0.6 for the best models) lower during the resuscitation phase compared to the blood loss phase (R2 > 0.8)? How do the authors explain this decrease in productivity, especially taking into account the assumptions made when calculating the true values of fluid balance (for example, 1:1 for whole blood, 3:1 for crystalloids)?
Thank you for your questions, the model training was done strictly in the hemorrhagic region of the study – the performance decrease seen is because the models are seeing data (resuscitation) that the model were not tested on. This was discussed as being intriguing due to PEBL still recovering in the resuscitation as though it had been trained on this data. Future work can involve retraining the AI models to include the resuscitation components, and we anticipate performance of the models would improve.
Round 2
Reviewer 1 Report
Comments and Suggestions for Authors
While the authors have responded many of the earlier issues thoroughly. However, still a few basic issues concerns remain unresolved and require further refinement before manuscript can be considered for acceptance:
- Citing the proof-of-concept nature of the study, the authors declined to conduct statistical comparisons such as paired t-tests or Wilcoxon signed-rank tests. Nonetheless, even with small datasets, basic statistical comparisons (even non-parametric) are common practice and would support model comparison claims. Therefore, clearly acknowledge this limitation in both Results and Discussion sections or include a statistical comparison between final model performances (e.g., XGB vs. RF).
- The authors agreed that generalizability is limited due to lack of external datasets. This is acceptable but clearly frame this as an internal proof-of-concept study in the abstract and conclusion (not just in the discussion).
- Despite improvements to axis labels and legends, several figures (e.g., Figures 3–6) remain difficult to understand due to multiple overlaid subpanels and inconsistent color coding. Figure 3 is duplicated in the manuscript, indicating a figure numbering error. This is a minor but essential change for clarity and consistency.
- Although learning curves were added in the supplement, but no summary or explanation is provided in the main text. Include a brief paragraph in the Results or Discussion to highlight any insights from these curves.
Author Response
While the authors have responded many of the earlier issues thoroughly. However, still a few basic issues concerns remain unresolved and require further refinement before manuscript can be considered for acceptance:
- Citing the proof-of-concept nature of the study, the authors declined to conduct statistical comparisons such as paired t-tests or Wilcoxon signed-rank tests. Nonetheless, even with small datasets, basic statistical comparisons (even non-parametric) are common practice and would support model comparison claims. Therefore, clearly acknowledge this limitation in both Results and Discussion sections or include a statistical comparison between final model performances (e.g., XGB vs. RF).
The initial reservations were related to treating the problem as a 2-way parameter problem, but we have taken the reviewers recommendation and have simplified the statistical analyses to the following. Goodness of fit data were consistently flagged as non-parametric due to lack of normal distribution so when comparing three or more variables, Friedmans test for repeated measures, post hoc Dunn’s tests were used when comparing the four models or the four sampling windows for each model. When comparing only two models, Wilcoxon matched pairs rank tests were used. As such, p-values have been added throughout and a number of figures have been added to better display statistical significance. We have further added the description of these methods in the methodology section. We hope these adjustments address this issue.
- The authors agreed that generalizability is limited due to lack of external datasets. This is acceptable but clearly frame this as an internal proof-of-concept study in the abstract and conclusion (not just in the discussion).
Good suggestion, these limitations about sample size and proof-of-concept have been added to the abstract and next steps in the conclusion.
- Despite improvements to axis labels and legends, several figures (e.g., Figures 3–6) remain difficult to understand due to multiple overlaid subpanels and inconsistent color coding. Figure 3 is duplicated in the manuscript, indicating a figure numbering error. This is a minor but essential change for clarity and consistency.
We have tried to address this but are not fully aware of the overlaid subpanel issue. However, we have fixed the Figure numbering and updated the coloring to make it more consistent across all figures. We hope this addresses this issue and improves clarity.
- Although learning curves were added in the supplement, but no summary or explanation is provided in the main text. Include a brief paragraph in the Results or Discussion to highlight any insights from these curves.
Results and discussion related to the learning curves have been added.
Reviewer 2 Report
Comments and Suggestions for Authors
The authors answered all the questions in detail and made the necessary adjustments to the article. The article is recommended for publication.
Author Response
Thanks for reviewing our paper.